# A Coating Based on Bioactive Compounds from *Streptomyces* spp. and Chitosan Oligomers to Control *Botrytis cinerea* Preserves the Quality and Improves the Shelf Life of Table Grapes

**DOI:** 10.3390/plants12030577

**Published:** 2023-01-28

**Authors:** Laura Buzón-Durán, Eva Sánchez-Hernández, Mercedes Sánchez-Báscones, Mari Cruz García-González, Salvador Hernández-Navarro, Adriana Correa-Guimarães, Pablo Martín-Ramos

**Affiliations:** 1Department of Agroforestry Sciences, ETSIIAA, Universidad de Valladolid, Avenida de Madrid 44, 34004 Palencia, Spain; 2Department of Agricultural and Forestry Engineering, ETSIIAA, Universidad de Valladolid, Avenida de Madrid 44, 34004 Palencia, Spain

**Keywords:** *Botrytis cinerea*, *Streptomyces* spp., postharvest diseases, synergism, antifungal, table grapes, conjugate complexes

## Abstract

*Botrytis cinerea* is the most harmful postharvest disease of table grapes. Among the strategies that can be envisaged for its control, the use of coatings based on natural products is particularly promising. The study presented herein focuses on the assessment of the antagonistic capacity of two *Streptomyces* species and their culture filtrates against *B. cinerea*. Firstly, the secondary metabolites were characterized by gas chromatography–mass spectrometry, with N1-(4-hydroxybutyl)-N3-methylguanidine acetate and 2R,3S-9-[1,3,4-trihydroxy-2-butoxymethyl]guanine acetate as the main compounds produced by *S. lavendofoliae* DSM 40217; and cyclo(leucyloprolyl) and cyclo(phenylalanylprolyl) as the most abundant chemical species for *S. rochei* DSM 41729. Subsequently, the capacity of *S. lavendofoliae* DSM 40217 and *S. rochei* DSM 41729 to inhibit the growth of the pathogen was tested in dual culture plate assays, finding 85–90% inhibition. In agar dilution tests, their culture filtrates resulted in effective concentration values (EC_90_) in the 246–3013 μg·mL^−1^ range. Upon the formation of conjugate complexes with chitosan oligomers (COS) to improve solubility and bioavailability, a synergistic behavior was observed, resulting in lower EC_90_ values, ranging from 201 to 953 μg·mL^−1^. Ex situ tests carried out on ‘Timpson’ and ‘Red Globe’ table grapes using the conjugate complexes as coatings were found to maintain the turgor of the grapes and delay the appearance of the pathogen by 10−15 days at concentrations in the 750−1000 µg·mL^−1^ range. Hence, the conjugate complexes of COS and the selected *Streptomyces* spp. culture filtrates may be put forward as promising protection treatments for the sustainable control of gray mold.

## 1. Introduction

Table grapes are considered a product with functional characteristics since they offer a wealth of health benefits due to their high nutrient and antioxidant contents [1], which are essential for a healthy human diet [2]. Table grapes are one of the most consumed fruits globally and are cultivated in about 90 countries [3]. Like other fresh fruits, grapes are very delicate and the losses at harvest and during distribution are very high [4], which makes them a highly perishable food with relatively short postharvest storage lives [2].

Postharvest diseases of table grapes in favorable conditions—and particularly in developing countries—can cause losses in total production of up to 55% [5]. This situation is aggravated by the fact that only 5% of all agricultural research funding is allocated to postharvest challenges, with the majority of investments going toward expanding crop production strategies [6]. However, compared to production strategies, reducing postharvest loss and waste is more time and money efficient. Additionally, reducing the amount of fresh fruit lost entails a decrease in the amount of land, chemicals, energy, and other inputs needed to produce horticultural crops, safeguarding the environment and preserving natural resources [7,8].

In the case of table grapes, gray mold, caused by *Botrytis* spp., is the most harmful postharvest disease in, given that it can grow and spread even at very low temperatures (0.5 °C). Infections frequently occur in the field, where they remain dormant until ripening [9]. To prolong the shelf life of table grapes against *B. cinerea*, different technologies have been used to minimize losses caused during handling, transport, and postharvest storage of table grapes, as summarized in the review by De Simone et al. [10]. Because of the capacity of *B. cinerea* to develop resistance swiftly, conventional synthetic fungicides, such as anilinopyrimidines, dicarboximides, hydroxyanilides, methyl benzimidazole carbamates, succinate dehydrogenase inhibitors, and quinone inhibitors are not a long-term solution [11]. An alternative strategy consists in using a coating material to wrap the berry, in particular edible coatings manufactured with natural polymers, compliant with food safety requirements [12]. These coating materials can increase the shelf-life of fruit crops and retain quality while lowering water losses [10].

Among the natural polymers that may be used in these coatings, chitosan has been widely studied for postharvest treatment in grapes due to its high antifungal potential [10]. However, studies on its synergistic effect with bioactive compounds are scarce.

A promising source of such bioactive compounds is actinobacteria. *Streptomyces* spp. produces over 7600 biological chemicals, about 7500 biologically active secondary metabolites, and about 75% of all antibiotics. As a result, *Streptomyces* have become the primary source of antibiotics utilized in drug discovery [13]. Members of the genus *Streptomyces* such as *S. rochei* can produce compounds with antibiotic properties (e.g., lankacidin [14] and streptothricin [15]), apoptosis inducers (e.g., ravidomycin analogs [16]), and antifungals (e.g., borrelidin [17], butyrolactol A, and butyrolactol B [18]). In turn, *S. lavendofoliae* produces antibiotics such as anthracidin A [19] and aclacinomycin; antifungals such as phosphazinomycins-hydrazides [20]; carboxypeptidase inhibitors (e.g., piperastatin A [21] and piperastatin B [22]), and compounds with antimicrobial and immunosuppressive activity (e.g., depsidomycin [23]).

In view of the wide repertoire of activities of the chemical species present in their secondary metabolites, the aim of this study was: (1) to identify by gas chromatography–mass spectrometry (GC–MS) the bioactive compounds with antifungal activity generated by *S. rochei* DSM 41729 and *S. lavendofoliae* DSM 40217; (2) to evaluate the in vitro antifungal activity against *B. cinerea* by *S. lavendofoliae* DSM 40217 and *S. rochei* DSM 41729 through dual plate assays and mycelial growth-inhibition tests involving their secondary metabolites, alone or in combination with chitosan oligomers (COS); and (3) to test ex situ the best treatments as table grape coatings.

## 2. Results

### 2.1. GC–MS Characterization of Secondary Metabolites of S. lavendofoliae DSM 40217 and S. rochei DSM 41729

The main secondary metabolites identified for *S. lavendofoliae* DSM 40217 (Figure 1) were N1-(4-hydroxybutyl)-N3-methylguanidine acetate (19.7%); 2R,3S-9-[1,3,4-trihydroxy-2-butoxymethyl]guanine acetate (7.8%); benzeneacetaldehyde (5.3%); hexahydro-3-(2-methylpropyl)-pyrrolo [1,2-a]pyrazine−1,4-dione (3.5%); N,N-dimethylglycine (2.3%); 2-(2-hydroxyethyl)piperidine (1.8%); 3(5)-[[1,2-dihydroxy-3-propoxy]methyl]-4-hydroxy−1H-pyrazole-5(3)-carboxamide (1.6%); glycerol 1,2-diacetate (1.5%); pentanal, 2-methylene-, 2-(1-methylethyl)hydrazone (1.5%); 1,2-cyclopentanedione (1.4%); dihydroergotamine (0.4%); butyrolactone (0.45%); and deoxyspergualin (0.4%). A complete list is presented in Appendix A.

Concerning *S. rochei* DSM 41729 extracts (Figure 2), the GC–MS profile included cyclo(leucyloprolyl) and cyclo(phenylalanylprolyl) (pyrrolo [1,2-a]pyrazine−1,4-dione, hexahydro-3-(2-methylpropyl)-; and pyrrolo [1,2-a]pyrazine−1,4-dione, hexahydro-3-(phenylmethyl)-, respectively; 11.6%); 2,3-butanediol (6.0); D-leucine (5.6%); di-[1,3,2]-oxazino [6,5-f:5′,6′-H]quinoxaline, 2,3,4,5,6,7-hexahydro-3,6-bis [2-diethylaminoethyl]−10,11-diphenyl- (4.9%); dianhydromannitol (4.2%); N,N-dimethylglycine (4.1%); 1,1-bis[aziridyltrimethyl amine] (3.6%); benzeneacetaldehyde (3.2%); isosorbide (2.0%); sebacic acid, 2,6-dimethoxy phenyl octyl ester (2.0%); dl-mevalonic acid lactone (1.8%); n-propargyloxycarbonyl-l-norvaline, tetradecyl ester (1.7%); 3-pyrrolidin-2-yl-propionic acid (1.4%); adenine (1.0%); 3,5-dimethoxy-phenol, (0.9%); and deoxyspergualin (0.7%). A comprehensive list is provided in Appendix A.

### 2.2. Antifungal Activity

#### 2.2.1. Antibiosis Assay

Dual plate assays showed a strong antagonistic effect of *S. rochei* DSM 41729 and *S. lavendofoliae* DSM 40217 against *B. cinerea*. Fungal growth was reduced by 80% in the case of *S. lavendofoliae* DSM 40217, and by 85% in the case of *S. rochei* DSM 41729 (Appendix A). Such results are in agreement with those previously reported by Ghanem et al. [24] for *S. rochei* and two other *Streptomyces* spp.

#### 2.2.2. In vitro Growth Inhibition Tests

The results of the mycelial growth inhibition tests are summarized in Figure 3. When tested alone, a higher efficacy of COS than of both secondary metabolites was observed: full inhibition was reached at 1500 μg·mL^−1^ for the former, whereas secondary metabolites reduced mycelial growth but did not reach complete inhibition at the highest assayed concentration. Upon conjugation of the secondary metabolites with COS, an enhancement in terms of efficacy was found in both cases: the conjugated COS–*S. lavendofoliae* DSM 40217 metabolites treatment reached full inhibition at 1000 μg·mL^−1^ and the COS–*S. rochei* DSM 41729 metabolites treatment did so at 750 μg·mL^−1^ (Appendix A).

Upon comparison of the effective concentrations (Table 1), differences in the efficacy of the treatments could be observed more clearly. The highest efficacy (i.e., the lowest EC_50_ and EC_90_ values) corresponded to the COS–*S. rochei* DSM 41729 metabolites conjugate complex, followed by the COS*–S. lavendofoliae* DSM 40217 metabolites conjugate complex, COS, *S. rochei* DSM 41729 metabolites, and *S. lavendofoliae* DSM 40217 metabolites.

Regarding the synergy factors (Table 2), a synergistic behavior was found between COS and the metabolites (SF values ≥ 1). The highest synergy factor was obtained for the EC_90_ of COS–*S. lavendofoliae* DSM 40217 (SF = 2.33).

#### 2.2.3. Ex Situ Growth Inhibition Tests

As shown in Figure 4, the two treatments were very effective, achieving a high inhibition of the fungus until the end of the trial. Disease incidences on both cultivars of table grapes were calculated on days 1, 5, 10, and 15 (Appendix A). In the negative controls, the fungus did not proliferate (discarding contamination), while in the clusters that had been artificially inoculated with the fungus, but which were not treated, *B. cinerea* managed to invade 100% of the cv. ‘Timpson’ berries and ca. 80% of the cv. ‘Red Globe’ berries after 15 days. Regarding the cv. ‘Timpson’ clusters treated with COS–*S. lavendofoliae* DSM 40217 conjugate complex, no colonization by the pathogen was observed after 15 days, while in the cv. ‘Red Globe’ grapes some colonization was observed after 10 days, reaching an incidence of ca. 20% after 15 days. On the other hand, in the tests carried out with COS–*S. rochei* DSM 41729 conjugate complex coating, 100% effectiveness against *B. cinerea* after 15 days was observed for both cultivars (and at a lower concentration).

It is worth noting that in both cases, the treatments resulted in better preservation of the firmness and turgor of the berries. Regarding the berries’ color, it was not affected in the case of the cv. ‘Red Globe’ clusters, but the COS–*S. rochei* DSM 41729 conjugate complex coating conferred a slightly yellowish hue to the cv. ‘Timpson’ berries after day 5 (an effect that was not observed in the clusters treated with COS–*S. lavendofoliae* DSM 40217 conjugate complex). This should be taken into consideration, as it may affect consumers’ preferences.

## 3. Discussion

### 3.1. On the Secondary Metabolites Profiles

Concerning the compounds identified in *S. lavendofoliae* DSM 40217, N1-(4-hydroxybutyl)-N3-methylguanidine acetate is related to the important fungicide named dodine (n-dodecylguanidine acetate) [25]. Hexahydro-3-(2-methylpropyl)-pyrrolo [1,2-a]pyrazine−1,4-dione (also found in *Eucalyptus urophylla* leaves extract [26]), is a member of the hexahydro pyrrolo [1,2-a]pyrazine−1,4-dione family, with known antimicrobial activity [27]. Pentanal, 2-methylene, 2-(1-methylethyl)hydrazone, and 2-propanone, dimethylhydrazone are hydrazones (analogs to hydrazides referred for *S. lavendofoliae* DSM 40217) that have potential antifungal activity [28]. Butyrolactone, as butyrolactol A and B referred to *S. rochei*, possesses antifungal activity [29].

Deoxyspergualin (also present in *S. rochei* DSM 41729) has been reported to promote resistance to *Candida albicans* (apart from being one of the best immunosuppressive, radiation-protective, antineoplastic, and hypoglycemic agents) [30]. Regarding the secondary metabolites identified in *S. rochei* DSM 41729, cyclo(leucyloprolyl) and cyclo(phenylalanyl-prolyl) regulate the growth of some fungi [31] and the latter one has been reported to exhibit activity against *C. albicans*. Oxazinoquinoxalines have also been reported to exhibit antibacterial activity [32].

### 3.2. Efficacy of the Treatments

In relation to the efficacy of the treatments, the review paper by De Simone et al. [10] provides a qualitative and quantitative comparison of different biological compounds—including chitosan—for the control *B. cinerea* in table grapes. However, specific inhibition rates and the associated concentrations/effective concentrations of *Streptomyces* spp.-based treatments were not covered. A survey of such values against the pathogen under study is summarized in Table 3 for comparison purposes.

Taking into consideration that both the *B. cinerea* isolates and the *Streptomyces* species/strains differ from one article to another, a word of caution seems necessary regarding the comparisons presented below.

Concerning the antibiosis assay, a comparison of the inhibition percentages obtained herein (80% for *S. lavendofoliae* DSM 40217 and 85% for *S. rochei* DSM 41729) with the other *Streptomyces* spp. in the table suggests that only *Streptomyces* sp. A3265, *S. globisporus* JK−1, and *Streptomyces* sp. S97 would have a higher antifungal potential against *B. cinerea* in dual plate assays.

Regarding the minimum inhibitory concentration when metabolites were tested, the EC_90_ values of 1723 and 3013 µg·mL^−1^ obtained for *S. rochei* DSM 41729 and *S. lavendofoliae* DSM 40217, respectively, indicate that total inhibition of pathogen growth would be achieved at a dose approximately ten times lower than that reported for *S. globisporus* JK−1 (30,000 µg·mL^−1^), one of the species that outperformed the species tested herein in the dual plate assays. Therefore, the active compounds present in the secondary metabolites of *S. rochei* DSM 41729 and *S. lavendofoliae* DSM 40217 may have more promising antifungal activity.

In relation to chitosan, it has been widely studied for the postharvest treatment of grapes. In some previous studies, chitosan failed to fully inhibit the radial growth of the fungus, reducing the mycelium growth to 3.84 cm, as compared to 8.4 cm for the control [40]. In the work by Ramos-Bell et al. [41], in which a concentration of 15,000 µg·mL^−1^ was tested (approximately ten times higher than the MIC obtained in our trials, 1500 µg·mL^−1^), a 93.38% inhibition was found. Muñoz et al. [42] obtained an EC_50_ of 1770 µg·mL^−1^, a concentration approximately equal to the EC_90_ of the assay. These differences may be attributed to the use of chitosan oligomers instead of medium or high molecular-weight chitosan, given that the former ones have a higher antifungal activity [43].

With the ex situ experiment, it has been shown that cv. ‘Red Globe’ is more resistant to *B. cinerea* than other grape varieties, such as ‘Italia’ or ‘Victoria’ [9]. These results would be in agreement with our findings, provided that cv. ‘Timpson’ was more sensitive to *B. cinerea* than cv. ‘Red Globe’. In terms of the appearance of the treated grapes vs. untreated control grapes, in the case of cv. ‘Red Globe’, the obtained results are similar to those reported by Xu et al. [40]. Although there was no fungal development in the grapes treated with the coatings, on day 10 the firmness and turgor began to decline. Given that the berries of this cultivar have thinner skin than those of other cultivars, it could be a possible explanation for the proliferation of the fungus [44]; microcracks make the skin more permeable, which speeds up transpiration and water absorption (depending on humidity and internal pressure in the berry), resulting in increased fruit deterioration and a decrease in fruit firmness [45].

### 3.3. Mechanism of Action

In relation to the antifungal activity of COS, it is well established in the literature and may be ascribed to several mechanisms of action on the fungal cell wall [46]. The presence of chitosan polysaccharides confers a positive molecular charge, which, upon interaction with the negatively charged fungal cell membrane (lipopolysaccharides, phospholipids, or lipoproteins) [47], leads to changes in the cell membrane, such as increased cell permeability [48]. In turn, this results in changes in its functionality, in addition to the loss of intracellular components such as proteins and Na^+^ and Ka^+^ ions, which affects the osmotic pressure and finally cause cell death [49]. It has also been hypothesized that COS can significantly alter cellular chitin levels and thereby disturb the equilibrium between degradation and synthesis, resulting in a weakening of the cell wall [50]. In addition, excessive ROS attacks intracellular biomolecules such as lipids, proteins, and DNA, causing irreversible oxidative damage and thereby triggering apoptosis and necrosis. Furthermore, COS can infiltrate into the nucleus of fungus and disrupt the synthesis of DNA, as well as RNA [51]. As for its effect on grapes, chitosan can significantly increase the enzymatic activities of grapefruit, including superoxide dismutase, peroxidase, catalase, and ascorbate peroxidase, reducing the damage by active oxygen [43]. Regarding its role in the conjugate complexes, the observed synergism may result from an improved additive fungicidal activity per se or by simultaneous action on various fungal metabolic sites [52], but it may also be due to the fact that chitosan oligomers can enhance the solubility and bioavailability of the bioactive compounds present in the secondary metabolite filtrate [53,54].

Regarding the mode of action of *S. rochei* DSM 41729 and *S. lavendofoliae* DSM 40217, volatile organic compounds (VOCs) produced by *Streptomyces* spp. have been previously shown to control postharvest diseases. For instance, the volatiles emitted from *S. setonii* WY228 effectively controlled the postharvest pathogen of sweet potato *Ceratocystis fimbriata* Ellis et Halsted, 1890 [55]. Volatiles mainly downregulated the ribosomal synthesis genes and activated the proteasome system of the fungus, while the genes for spore development, cell membrane synthesis, mitochondrial function, and hydrolase and toxin synthesis were also downregulated, suggesting that VOCs have a variety of mechanisms of action on fungal pathogens [56].

Concerning other grapevine diseases, secondary metabolites from *S. rochei* DSM 41729 and *S. lavendofoliae* DSM 40217 have also been demonstrated to have antifungal activity against downy mildew [57]. Such activity would arise from the presence of bioactive substances with antifungal capacity, as discussed in the previous section. However, further research would be needed to gain insight into the mechanisms of action of each of them and into the differences in terms of effectiveness depending on whether they act together or separately.

## 4. Materials and Methods

### 4.1. Bacterial and Fungal Isolates, Reagents, and Table Grapes

The two bacteria of the genus *Streptomyces*, viz. *Streptomyces rochei* (DSM 41729) and *Streptomyces lavendofoliae* (DSM 40217), were purchased from DSMZ (Deutsche Sammlung von Mikroorganismen und Zellkulturen; Braunschweig, Germany). The *Botrytis cinerea* (CECT 20973) isolate came from CECT (Spanish Type Culture Collection, Valencia, Spain).

High molecular weight chitosan, (CAS 9012-76-4; 310,000-375,000 Da), was purchased from Hangzhou Simit Chemical Technology Co., Ltd. (Hangzhou, China). Phosphate buffer (for microbiology, APHA, pH 7.2), ethyl acetate (CAS 141-78-6), and citric acid (CAS 77-92-9) were supplied by Sigma-Aldrich Química S.A. (Madrid, Spain). Neutrase^®^ 0.8 L enzyme was supplied by Novozymes (Bagsvaerd, Denmark). Potato dextrose agar (PDA), potato dextrose broth (PDB), yeast extract, and Bacto^TM^ Peptone were purchased from Becton, Dickinson and Company (Franklin Lakes, NJ, USA). Tryptone soy broth (TSB), starch casein agar (SCA), Mueller–Hinton agar, and malt extract agar (MEA) came from Oxoid Ltd. (Hampshire, UK). Molasses was supplied by ACOR, Sociedad Cooperativa General Agropecuaria (Castilla y León, Spain).

The organic farming cv. ‘Red Globe’ and ‘Timpson’ grapes used in ex situ assays were supplied by FRUAMO, Las Cabezuelas Sociedad Cooperativa (Murcia, Spain).

### 4.2. Preparation of Secondary Metabolites of Streptomyces spp. and Preparation of B. cinerea Conidial Suspension

Lyophilized *S. lavendofoliae* (DSM 40217) and *S. rochei* (DSM 41729) were inoculated in TSB at 28 °C for 24 h and were seeded on SCA medium plates at 28 °C for 10 days. The plates were stored at 4 °C. Secondary metabolites were obtained following the method described by Sadigh-Eteghad et al. [58]. After fermentation, each final solution of the two cultures was treated with 50 mL of phosphate buffer and then sonicated for 5 min (pH 6.4). After centrifuging the filtrates, the bioactive compound-containing supernatant was extracted with 100 mL of ethyl acetate, concentrated under pressure, and then freeze-dried. In agreement with Pazhanimurugan et al. [59], the culture filtrates had a concentration of approx. 2000 µg·mL^−1^ (1958 µg·mL^−1^ for *S. lavendofoliae* DSM 40217 secondary metabolites and 1877 µg·mL^−1^ for *S. rochei* DSM 41729 secondary metabolites). For in vitro and ex situ antifungal activity tests, the freeze-dried samples were dissolved in Milli-Q^®^ water.

For the preparation of *B. cinerea* conidial suspension, lyophilized *B. cinerea* (CECT 20973) was inoculated in PSB at 26 °C for 24 h and was seeded on PDA medium plates at 26 °C for 7 days. Subsequently, 5 plugs (10 mm in diameter) were inoculated in 150 mL of PDB. Seven-day-old PDB cultures (150 mL cultures kept at 26 °C, 135 rpm in an orbital stirrer incubator) were harvested for *B. cinerea* conidia. The suspension was filtered twice to eliminate hyphal fragments before measuring the spore concentration using a hemocytometer (Weber Scientific International Ltd., Teddington, Middlesex, UK). It was then amended with 0.2% Tween 20^®^ and adjusted to a final concentration of 1 × 10^6^ spores (conidia)·mL^−1^.

### 4.3. Synthesis of Chitosan Oligomers and COS–Secondary Metabolites Conjugate Complexes

The procedure reported by Santos-Moriano et al. [60] with some modifications was used to obtain chitosan oligomers. 20 g of chitosan was stirred with 20 g of citric acid in T1 L of Milli-Q^®^ water at 50 °C until a homogeneous solution was obtained. Then, the enzyme Neutrase^®^ 0.8 L (1.67 g·L^−1^) was added to degrade the polymer chains. Subsequently, the mixture was subjected to sonication in 3 min cycles with 1 min stops in between. Once the process was completed, oligomers of molecular weight < 2000 Da were obtained in a solution with a pH of 4–6. The molar mass of the COS samples was determined by measuring the viscosity, in agreement with Yang et al. [61] in a solvent of 0.20 mol·L^−1^ NaCl + 0.1 mol·L^−1^ CH_3_COOH at 25 °C using an Ubbelohde capillary viscometer. Molar mass was determined using the Mark–Houwink equation [*η*] = 1.81 × 10^−3^ M^0.93^ [62].

Conjugate complexes of secondary metabolites of *S. rochei* DSM 41729 or *S. lavendofoliae* DSM 40217 and COS were obtained according to the procedure previously described in [63], by mixing in a 1:1 ratio (*w/w*: 2000 µg·mL^−1^/2000 µg·mL^−1^), followed by sonication. The resulting stock solution (at a concentration of 2000 µg·mL^−1^) was diluted in Milli-Q^®^ water to obtain the different concentrations assayed in the mycelial growth inhibition tests.

### 4.4. Antifungal Activity Assessment

#### 4.4.1. In Vitro Tests of Mycelial Growth Inhibition

For the antagonistic study of *B. cinerea* versus both *Streptomyces* spp., the methodology of Ghanem et al. [24] was used, with slight modifications. PDA was used as the culture medium, in which 4 drops of 30 µL from TSB tubes previously seeded at 26 °C for 18 h were added to reach a concentration of 1 × 10^8^ CFU·mL^−1^. The plates were then incubated at 26 °C for 3 days. Then, a 5 mm plug of *B. cinerea* was added in the center while another plug was inoculated on a control plate. Both were incubated for 7 days at 26 °C. The experiment was performed in triplicate with three replicates. The growth inhibitory effects on the pathogenic fungi were represented as a percentage of pathogen growth inhibition (PGI) and calculated according to Zambrano et al. [64], using the following equation: PGI = (R_1_ − R_2_)/R_1_ × 100, where R_1_ = growth of pathogenic fungus in the control plate and R_2_ = growth of pathogenic fungus interacting with the antagonist.

For the mycelial growth inhibition assays with the five treatments (viz. COS, *S. lavendofoliae* DSM 40217 secondary metabolites, *S. rochei* DSM 41729 secondary metabolites, COS–*S. lavendofoliae* DSM 40217 secondary metabolites, and COS–*S. rochei* DSM 41729 secondary metabolites), the methodology reported by Sánchez-Hernández et al. [65] was chosen. PDA plates with 10 concentrations of each treatment (ranging from 62.5 to 1500 µg·mL^−1^) were inoculated with 5 mm diameter plugs, and incubated at 26 °C for 7 days, together with the control plate, containing only culture medium. Growth inhibition was calculated according to the formula: ((d_c_ − d_t_)/d_c_) × 100, where d_c_ is the average colony diameter in the control and d_t_ is the average diameter of the treated colony. EC_50_ and EC_90_ (50% and 90% effective concentrations, respectively) were estimated using PROBIT analysis in IBM SPSS Statistics v.25 software (IBM; Armonk, NY, USA). The level of interaction, i.e., the synergy factor (SF), was estimated according to Wadley’s method [66].

#### 4.4.2. Ex Situ Tests of Mycelial Growth Inhibition

The effectiveness of the different conjugate complex-based coatings on gray mold incidence was also assayed ex situ on cv. ‘Red Globe’ and ‘Timpson’ table grapes. The MIC values obtained in the in vitro mycelial growth inhibition assays were used, viz. 750 and 1000 µg·mL^−1^ for COS–*S. rochei* DSM 41729 secondary metabolites and COS–*S. lavendofoliae* DSM 40217 secondary metabolites, respectively.

Two cultivars of organic table grapes were tested: green seedless cv. ‘Timpson’ grapes, which are more sensitive to *B. cinerea*, and cv. ‘Red Globe’ red grapes with seeds, which are more tolerant to *B. cinerea*. Once harvested, the grapes were sent from Murcia to the University of Valladolid facilities by an express courier service at 4 °C. The trial began 24 h after the grapes had been harvested.

Forty bunches of uniform size, color, and shape were selected to obtain homogeneous lots, all of which were checked to ensure that they were free of lesions. The trials were carried out in triplicate (120 bunches). At the start of the trials, grapes were superficially disinfected with a 3% NaOCl solution for 2 min, washed three times with sterile distilled water, and dried in a laminar flow hood using sterile absorbent paper. In this state, the negative control grapes (i.e., no fungus and no treatment) were reserved. Subsequently, the grapes were immersed in the respective treatments (viz. COS–*S. lavendofoliae* DSM 40217 and COS–*S. rochei* DSM 41729 secondary metabolites) for 5 min. Once dry, 2 mm cuts were made with a scalpel on the grapes, which were then inoculated with 10 µL of a conidia suspension (10^6^ conidia mL^−1^) [67]. Some untreated grapes were reserved to make cuts and add the conidia, using them as positive controls (with fungus and without treatment). The grapes were then stored in a chamber at 22 °C with a relative humidity of 90%, i.e., under conditions suitable for the development of *B. cinerea*. The trial lasted 15 days, the time it took for the fungus to completely colonize the grapes of the positive control. During this time, the quality of the grapes was studied in terms of morphological, color, and textural changes, together with the time of appearance and colonization of the fungus.

### 4.5. Gas Chromatography–Mass Spectrometry Analysis of Streptomyces spp. Secondary Metabolites

Part of the culture filtrates was reserved to identify the active compounds produced by *S. rochei* DSM 41729 and *S. lavendofoliae* DSM 40217. For this purpose, the culture filtrates were centrifuged and the supernatant, which contained the bioactive compounds, was extracted with 100 mL of ethyl acetate, concentrated under pressure, and freeze-dried. Then, 25 mg of the obtained freeze-dried aqueous extract was dissolved in 5 mL of HPLC-grade methanol to obtain a 5 mg·mL^−1^ solution, which was filtered and used for GC–MS analysis. The GC–MS analysis was carried out at the Research Support Services (STI) at Universidad de Alicante (Alicante, Spain), using a gas chromatograph model 7890A coupled to a quadrupole mass spectrometer model 5975C (both from Agilent Technologies, Santa Clara, CA, USA). Chromatographic conditions: 1 µL injection volume; 280 °C injector temperature, in splitless mode; initial oven temperature = 60 °C, 2 min, followed by a ramp of 10 °C·min^−1^ up to a final temperature of 300 °C, 15 min. The chromatographic column used for the separation of the compounds was an Agilent Technologies HP-5MS UI of 30 m in length, 0.250 mm diameter, and 0.25 µm film. Mass spectrometer conditions: temperature of the electron impact source of the mass spectrometer = 230 °C and of the quadrupole = 150 °C; ionization energy = 70 eV. The identification of components was based on a comparison of their mass spectra and retention time with those of the authentic compounds and by computer matching with the database of the National Institute of Standards and Technology (NIST11).

### 4.6. Statistical Analysis

Data were subjected to analysis of variance (ANOVA) in IBM SPSS Statistics v.25 software. Tukey’s HSD test at 0.05 probability level (*p* < 0.05) was used for the post hoc comparison of means.

## 5. Conclusions

Gas chromatography–mass spectrometry analyses evidenced that *S. rochei* DSM 41729 and *S. lavendofoliae* DSM 40217 produce bioactive compounds with antimicrobial properties, including N1-(4-hydroxybutyl)-N3-methylguanidine acetate; hexahydro-3-(2-methylpropyl)-pyrrolo [1,2-a]pyrazine−1,4-dione; hydrazones; butyrolactone; deoxyspergualin; cyclo(leucyloprolyl); cyclo(phenylalanyl-prolyl); and oxazinoquinoxalines. In the in vitro mycelial growth inhibition tests, the presence of these chemical species resulted in EC_90_ values in the 246–3013 μg·mL^−1^ range, which were substantially improved (EC_90_ = 201–953 μg·mL^−1^) upon the formation of conjugate complexes with COS. Ex situ bioassays showed that the COS–*S. rochei* DSM 41729 secondary metabolites conjugate complex at a concentration of 750 μg·mL^−1^ fully inhibited the growth of *B. cinerea* on cv. ‘Timpson’ and cv. ‘Red Globe’ grapes for 15 days. Regarding the clusters treated with COS–*S. lavendofoliae* DSM 40217 secondary metabolites conjugate complex at 1000 μg·mL^−1^, full protection was observed for cv. ‘Timpson’ berries, while an incidence of ca. 20% was registered in cv. ‘Red Globe’ grapes after 15 days. These promising results support that coatings based on secondary metabolites of *S. rochei* DSM 41729 and *S. lavendofoliae* DSM 40217 can be a promising strategy to control *B. cinerea,* preserve the quality, and improve the shelf life of table grapes.

## Figures and Tables

**Figure 1 plants-12-00577-f001:**
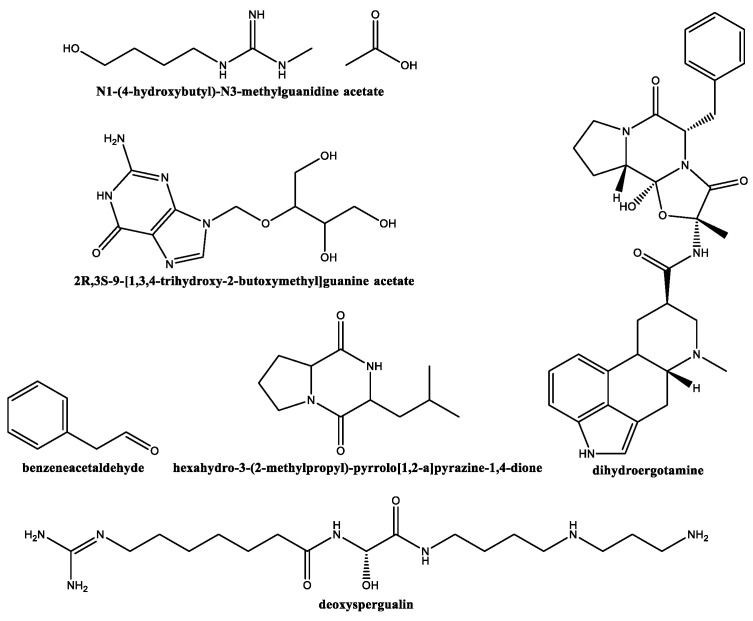
Compounds identified by GC–MS in *S. lavendofoliae* DSM 40217 extract.

**Figure 2 plants-12-00577-f002:**
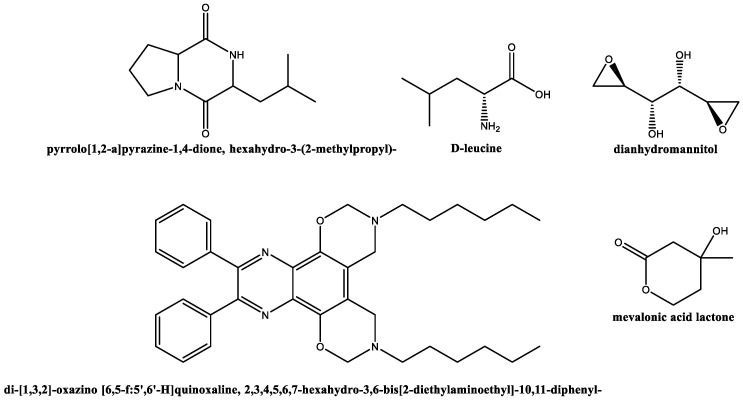
Compounds identified by GC–MS in *S. rochei* DSM 41729 extract.

**Figure 3 plants-12-00577-f003:**
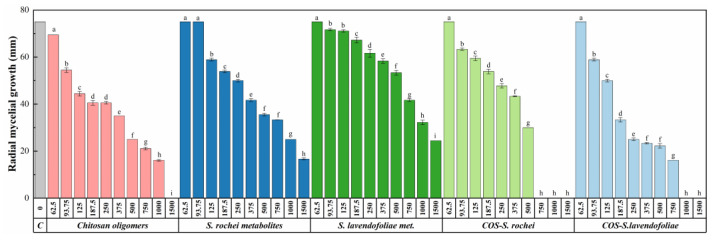
Radial growth values of *B. cinerea* in the presence of the different treatments under study at different concentrations (in μg·mL^−1^). Concentrations labeled with the same uppercase letters are not significantly different at *p* < 0.05 by Tukey’s test. All values are presented as the average of three repetitions. Error bars represent the standard deviation across three replicates.

**Figure 4 plants-12-00577-f004:**
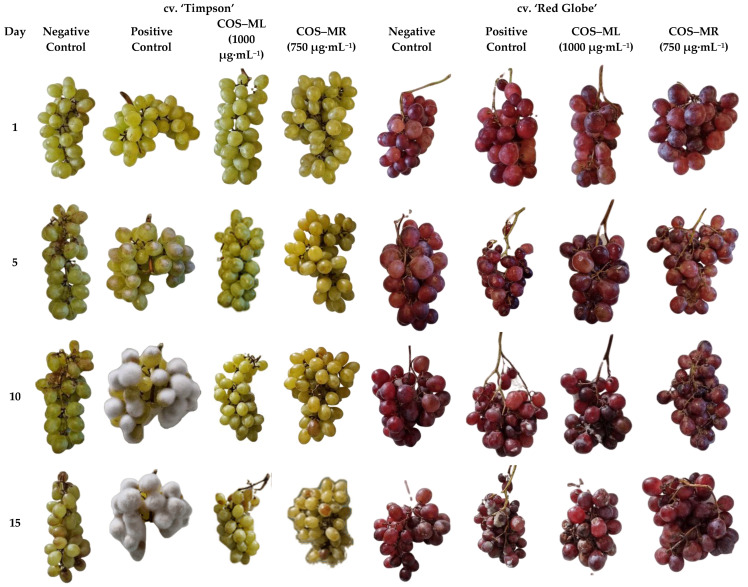
General appearance of cv. ‘Timpson’ and cv. ‘Red Globe’ clusters on days 1, 5, 10, and 15. From left to right: negative control; positive control (clusters inoculated with 1 × 10^4^ CFU·mL^−1^ of *B. cinerea*); clusters treated with COS–*S. lavendofoliae* DSM 40217 metabolites conjugate complex (COS–ML) at 1000 μg·mL^−1^ and inoculated with 1 × 10^4^ CFU·mL^−1^ of *B. cinerea;* clusters treated with COS–*S. rochei* DSM 41729 metabolites conjugate complex (COS–MR) at 750 μg·mL^−1^ and inoculated with 1 × 10^4^ CFU·mL^−1^ of *B. cinerea*. Only one replicate per treatment per day is shown.

**Table 1 plants-12-00577-t001:** EC_50_ and EC_90_ effective concentrations, expressed in µg·mL^−1^.

Effective Concentration	COS	*S. rochei* DSM 41729 Metabolites	*S. lavendofoliae* DSM 40217 Metabolites	COS–*S. rochei* DSM 41729 Metabolites	COS–*S. lavendofoliae* DSM 40217 Metabolites
EC_50_	246	429	909	201	311
EC_90_	1422	1723	3013	721	953

EC_50_: Effective concentration to reduce mycelial growth by 50%. EC_90_: Effective concentration to reduce mycelial growth by 90%.

**Table 2 plants-12-00577-t002:** Synergy factors for each of the conjugate complexes against *B. cinerea*.

Effective Concentration	Synergy Factor	
	COS–*S. rochei* DSM 41729 Metabolites	COS–*S. lavendofoliae* DSM 40217 Metabolites
EC_50_	1.55	1.24
EC_90_	1.89	2.33

EC_50_: Effective concentration to reduce mycelial growth by 50%. EC_90_: Effective concentration to reduce mycelial growth by 90%.

**Table 3 plants-12-00577-t003:** Concentration and associated inhibition rates reported in the literature for other *Streptomyces* species-based treatments against *B. cinerea*.

Streptomyces ssp.	Provenance of Isolate	In Vitro Effectiveness	In Vivo Assays		Ref.
			Fruit	Effectiveness	
*Streptomyces* sp. sdu1201	China	IR = 78.05%	Strawberry fruits cv. ‘Tian Bao’	CE = 53.33%, after 2 days	[33]
				CE = 45.44%, after 3 days	
*Streptomyces* sp. A3265	n.s.	MIC = 2.5–20 µg·mL^−1^	n.s.	n.s.	[34]
*S. globisporus* JK−1	China	IR = 100% at 30,000 µg·mL^−1^	Tomato fruits cv. ‘Annie’	DI = 35.8% after 24 h, 240,000 µg·mL^−1^	[35]
*S. nobilis* C51	China	IZ = 11.07	Tomato leaves	CE = 72.63%	[36]
*Streptomyces* sp. S97	Tunisia	IR = 99.3%	Strawberry fruits	DI = 87%	[37]
*S. griseus* MT210913”DG5”	Egypt	IR = 70.33%	n.s.		[24]
*S. rochei* MN700192”DG4”		IR = 70.83%			
*S. sampsonii* MN700191”DG1”		IR = 73.67%			
*S. rectiviolaceus* DY46	n.s.	n.s.	Tomato fruits	DI = 20%, at 100,000 µg·mL^−1^	[38]
*S. philanthi* RM−1−138	Thailand	IR = 73−100%	Tomato leaves	CE = 60%	[39]
			Tomato plants	CE = 57%	

IR = inhibition rate (%); CE = control effectiveness (%); n.s. = not specified; n.a. = no activity; DI = disease incidence (%); IZ = inhibition zone diameter (mm).

## Data Availability

The data presented in this study are available on request from the corresponding author.

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
