# Peer review of "A Coating Based on Bioactive Compounds from Streptomyces spp. and Chitosan Oligomers to Control Botrytis cinerea Preserves the Quality and Improves the Shelf Life of Table Grapes"

_plants, 2023, doi:10.3390/plants12030577_

Round 1

Reviewer 1 Report

An interesting and not very common topic is addressed in the area of control of postharvest pathogenic fungi by alternative methods to the use of pesticides, such as the use of Bioactive Compounds from Streptomyces spp. and Chitosan Oligomers to Control Botrytis cinerea , in the fruit of economic importance such as grapes. The work is well written, has a well-explained methodology, and adequate results are shown in figures and tables. I think it is a good manuscript.

Author Response

An interesting and not very common topic is addressed in the area of control of postharvest pathogenic fungi by alternative methods to the use of pesticides, such as the use of Bioactive Compounds from Streptomyces spp. and Chitosan Oligomers to Control Botrytis cinerea , in the fruit of economic importance such as grapes. The work is well written, has a well-explained methodology, and adequate results are shown in figures and tables. I think it is a good manuscript.

Response: Thank you for your positive feedback. It is most appreciated.

Reviewer 2 Report

The aim of this study is to identify the antifungal activity of the secondary metabolites from two strains of Streptomyces spp., alone or in combination with chitosan oligomers. The study design is acceptable. The study contains some valuable results and need suitable revisions.

1. Different strains from S. lavendofoliae and S. rochei may have different major metabolite components and antifungal activities. Therefore, the strain number of S. lavendofoliae and S. rochei should be added throughout the manuscript.

2. How to prepare the conjugated COS‐S. lavendofoliae metabolites, for instance with a final concentration of 1000 μg/mL? 2000 μg/mL of COS and S. lavendofoliae metabolites mixed with the ration of 1:1? Please described more detail in the section of method and materials.

3. Line 307, the secondary metabolites was extracted with ethyl acetate followed by freeze-dried. Which solvent used to dissolve the dried samples?

4. Line 264, ‘…chitosan enhances the bio-solubility of the bioactive compounds present in the secondary metabolite filtrate’. Please provide the reference to support this idea.

5. Line 328, ‘…, oligomers of molecular weight < 2000 Da were obtained…’. Method used to determine the molecular weight should be given.

Author Response

The aim of this study is to identify the antifungal activity of the secondary metabolites from two strains of Streptomyces spp., alone or in combination with chitosan oligomers. The study design is acceptable. The study contains some valuable results and need suitable revisions.

Q1. Different strains from S. lavendofoliae and S. rochei may have different major metabolite components and antifungal activities. Therefore, the strain number of S. lavendofoliae and S. rochei should be added throughout the manuscript.

Response: We thank the Reviewer for pointing this out. The strain numbers have been indicated throughout the manuscript every time that S. lavendofoliae DSM 40217 and S. rochei DSM 41729 are mentioned.

Q2. How to prepare the conjugated COS‐S. lavendofoliae metabolites, for instance with a final concentration of 1000 μg/mL? 2000 μg/mL of COS and S. lavendofoliae metabolites mixed with the ration of 1:1? Please described more detail in the section of method and materials.

Response: Given that the culture filtrates had a concentration of approx. 2000 µg·mL−1 (as explained in subsection 4.2), they were mixed with COS at a concentration of 2000 µg·mL−1, i.e., 1:1 w/w. The resulting stock solution (at a 2000 µg·mL−1) was diluted in Milli-Q® to obtain the various concentrations (in the 62.5 to 1,500 µg·mL−1 range) assayed in the in vitro tests of mycelial growth inhibition. The sentence at the end of section 4.3 has been rewritten to clarify this point, and now reads: “[…] The resulting stock solution (at a concentration of 2000 µg·mL−1) was diluted in Milli-Q® water to obtain the different concentrations assayed in the mycelial growth inhibition tests.

Q3. Line 307, the secondary metabolites was extracted with ethyl acetate followed by freeze-dried. Which solvent used to dissolve the dried samples?

Response: For in vitro and ex situ antifungal activity tests, the freeze-dried samples were dissolved in Milli-Q® water. A clarification has been included at the end of the first paragraph in subsection 4.2. Please kindly note that for GC-MS characterization HPLC grade methanol was used instead (as indicated in subsection 4.5).

Q4. Line 264, ‘…chitosan enhances the bio-solubility of the bioactive compounds present in the secondary metabolite filtrate’. Please provide the reference to support this idea.

Response: Two references have been added to support the statement, which has been slightly rephrased to specify that it is only applicable to chitosan oligomers (not to high-MW chitosan). It now reads: “[…] that chitosan oligomers can enhance the solubility and bio-availability of the bioactive compounds present in the secondary metabolite filtrate [10.3390/md13085156; 10.1016/j.ijbiomac.2015.02.039]

Q5. Line 328, ‘…, oligomers of molecular weight < 2000 Da were obtained…’. Method used to determine the molecular weight should be given.

Response: Subsection 4.3 has been updated to clarify that: “The molar mass of the COS samples was determined by measuring the viscosity, in agreement with Yang et al. [10.1007/s00217-005-0028-8] in a solvent of 0.20 mol·L−1 NaCl + 0.1 mol·L−1 CH3COOH at 25 °C using an Ubbelohde capillary viscometer. Molar mass was determined using the Mark–Houwink equation [η] = 1.81 × 10−3 M0.93 [10.1002/macp.1988.021890118].

Reviewer 3 Report

The study assesses the antagonistic capacity of two Streptomyces species and their culture filtrates against B. cinerea on table grapes which are highly perishable food with relatively short post‐harvest storage. The study is relevant since the aim is to consume various food products for as long as possible to reduce food loss and waste. In addition, Botrytis spp. is the most harmful post‐harvest disease; therefore, such research is highly appreciated. The study's strength is that several Streptomyces and grape species were investigated, allowing us to get a broader understanding.

The Introduction is well-written, and the problem s well as the objectives of the study are well-presented.

The results presented the secondary compounds of S. lavendofoliae and S. rochei determined by GC-MS. Moreover, the antifungal activity and mycelial growth inhibition are well presented in the graphs. The graphs and figures are high quality and resolution.

In the Discussion, the authors compared their results with previous studies, and a mini-review of other authors is presented. In addition, the authors gave attention to the mechanism of action.

In M&M, the Streptomyces bacteria genus and Botrytis cinerea isolate and the preparation of their conidial suspension, the evaluation of antifungal activity, and GC-MS analyses are presented. The experiment was performed in triplicate with three replicates, and the statistic analysis is described.

The Conclusions are written based on the results. However, it is not clearly summarized that the conclusions are common for both grape species.

Author Response

The study assesses the antagonistic capacity of two Streptomyces species and their culture filtrates against B. cinerea on table grapes which are highly perishable food with relatively short post‐harvest storage. The study is relevant since the aim is to consume various food products for as long as possible to reduce food loss and waste. In addition, Botrytis spp. is the most harmful post‐harvest disease; therefore, such research is highly appreciated. The study's strength is that several Streptomyces and grape species were investigated, allowing us to get a broader understanding.

The Introduction is well-written, and the problems well as the objectives of the study are well-presented.

The results presented the secondary compounds of S. lavendofoliae and S. rochei determined by GC-MS. Moreover, the antifungal activity and mycelial growth inhibition are well presented in the graphs. The graphs and figures are high quality and resolution.

In the Discussion, the authors compared their results with previous studies, and a mini-review of other authors is presented. In addition, the authors gave attention to the mechanism of action.

In M&M, the Streptomyces bacteria genus and Botrytis cinerea isolate and the preparation of their conidial suspension, the evaluation of antifungal activity, and GC-MS analyses are presented. The experiment was performed in triplicate with three replicates, and the statistic analysis is described.

Response: Thank you for your positive feedback on each of the aforementioned sections of the manuscript.

Q1. The Conclusions are written based on the results. However, it is not clearly summarized that the conclusions are common for both grape species.

Response: The conclusions have been expanded to specify the efficacy of the COS-Streptomyces spp. secondary metabolites conjugate complexes on each table grape cultivar, following the Reviewer’s suggestion: “[…] Ex situ bioassays showed that the COS-S. rochei DSM 41729 secondary metabolites conjugate complex at a concentration of 750 μg·mL−1 fully inhibited the growth of B. cinerea on cv. ‘Timpson’ and cv. ‘Red Globe’ grapes for 15 days. Regarding the clusters treated with COS-S. lavendofoliae DSM 40217 secondary metabolites conjugate complex at 1000 μg·mL−1, full protection was observed for cv. ‘Timpson’ berries, while an incidence of ca. 20% was registered in cv. ‘Red Globe’ grapes after 15 days. […]”